# Characterization of Chromatin Accessibility in Fetal Bovine Chondrocytes

**DOI:** 10.3390/ani13111875

**Published:** 2023-06-05

**Authors:** Qi Zhang, Qian Li, Yahui Wang, Yapeng Zhang, Ruiqi Peng, Zezhao Wang, Bo Zhu, Lingyang Xu, Xue Gao, Yan Chen, Huijiang Gao, Junwei Hu, Cong Qian, Minghao Ma, Rui Duan, Junya Li, Lupei Zhang

**Affiliations:** 1Key Laboratory of Animal (Poultry) Genetics Breeding and Reproduction, Ministry of Agriculture and Rural Affairs, Institute of Animal Science, Chinese Academy of Agricultural Sciences, Beijing 100193, China; esther_zq@163.com (Q.Z.); lq798711247@163.com (Q.L.); wang1434243198@163.com (Y.W.); 13592563258@163.com (Y.Z.); 82101215382@caas.cn (R.P.); wangzezhao@caas.cn (Z.W.); zhubo525@126.com (B.Z.); xulingyang@caas.cn (L.X.); gaoxue76@126.com (X.G.); chenyan0204@163.com (Y.C.); gaohuijiang@caas.cn (H.G.); 2Academy of Pingliang Red Cattle, 492 South Ring Road, Kongtong District, Pingliang 744000, China; hujw2@126.com (J.H.); m18793361772_1@163.com (C.Q.); mmhshuo@163.com (M.M.); z1256535304@163.com (R.D.)

**Keywords:** ATAC-seq, bovine fetal chondrocytes, transcription factors

## Abstract

**Simple Summary:**

Cartilage development plays a crucial role in human health and animal agriculture, and fetal development takes on particular importance in this process. In this study, we identified open chromatin regions in bovine fetal chondrocytes and peaks enriched in the promoter region. By integrating chromatin accessibility with transcriptome data and Genome-Wide Association Studies results of bovine stature traits, we further described the regulatory functions of these regulatory elements defined by open chromatin accessibility. Overall, these findings provide valuable information for understanding the regulatory mechanisms in cartilage development and conducting beef cattle genetic improvement programs.

**Abstract:**

Despite significant advances of the bovine epigenome investigation, new evidence for the epigenetic basis of fetal cartilage development remains lacking. In this study, the chondrocytes were isolated from long bone tissues of bovine fetuses at 90 days. The Assay for Transposase-Accessible Chromatin with high throughput sequencing (ATAC-seq) and transcriptome sequencing (RNA-seq) were used to characterize gene expression and chromatin accessibility profile in bovine chondrocytes. A total of 9686 open chromatin regions in bovine fetal chondrocytes were identified and 45% of the peaks were enriched in the promoter regions. Then, all peaks were annotated to the nearest gene for Gene Ontology (GO) and Kyoto Encylopaedia of Genes and Genomes (KEGG) analysis. Growth and development-related processes such as amide biosynthesis process (GO: 0043604) and translation regulation (GO: 006417) were enriched in the GO analysis. The KEGG analysis enriched endoplasmic reticulum protein processing signal pathway, TGF-β signaling pathway and cell cycle pathway, which are closely related to protein synthesis and processing during cell proliferation. Active transcription factors (TFs) were enriched by ATAC-seq, and were fully verified with gene expression levels obtained by RNA-seq. Among the top50 TFs from footprint analysis, known or potential cartilage development-related transcription factors FOS, FOSL2 and NFY were found. Overall, our data provide a theoretical basis for further determining the regulatory mechanism of cartilage development in bovine.

## 1. Introduction

Stature, body weight, meat quality, reproductive traits, etc., are important economic traits for cattle [1]. Body shape is directly related to economic traits such as carcass weight and pure meat percentage [2]. The biological basis of bovine body shape is mainly divided into skeletal development and muscle morphology. As a part of skeletal development, the long bones of the limb are the key factor determining the body height. The formation of long bones in mammals belongs to endochondral osteogenesis [3]. The specific process is that the mesoderm cells first proliferate and migrate to the developmental part of the bone in the embryo to form mesenchymal aggregation, and the mesenchymal cells differentiate into chondrocytes and secrete cartilage matrix [4]. At this time, the new differentiation occurs. The cells surrounding the chondrocytes form the perichondrium, which defines the boundaries of a developing bone. Chondrocytes differentiate into primary ossification centers and undergo chondrocyte hypertrophy, and then osteoblasts form cancellous bone, completing the longitudinal growth of long bones [5].

The process of chondrocyte proliferation involves complex regulatory networks such as paracrine, autocrine, and endocrine, and it is accompanied by the transcription and translation of a large amount of genetic information. TFs specifically bind cis-regulatory elements, such as enhancers and promoters, to regulate gene transcription. The majority of TFs bind to open chromatin surveyed from the ENCODE project and chromatin accessibility status reflects TF binding information [6]. It has been established that crucial genes involved in skeletal development constitute a regulatory network that jointly promotes endochondral ossification [7,8]. *SOX9* and *RUNX2* genes were highly expressed in chondrocyte chondrocytes. Transcriptional process of *Sox9* regulation Class I sites were mainly concentrated near the transcription start sites of highly expressed genes with non-chondrocyte-specific markers [9]. The enhancer in the intron of the *COL2A1* gene was initiated by the activation of the Sox family transcription factors to drive the expression of *COL2A1* [10]. The processes in which the *ACAN* gene is involved are highly conserved in various chondrocytes [11]. Sox family transcription factors are remotely regulated by upstream enhancer elements, making *ACAN* abundantly expressed in the initial stage of chondrocyte proliferation [12]. These transcription factors activate specific genes involved in the chondrocyte differentiation process and control cartilage formation in vivo.

The developing mammalian cartilage is mainly divided into resting zone, proliferative zone and hypertrophic zone. Chondrogenesis is the result of aggregation of mesenchymal cells. When mesenchymal cells aggregate and develop into chondrocytes, collagen types I, III, and V are expressed, while the differentiation of chondrogenic progenitor cells is accompanied by the specific expression of collagen types II, IX, and XI in cartilage [13]. After perichondrium formation, border-defining cells exit the cell cycle and undergo hypertrophy. Then, osteoblasts differentiate from the perichondrium and synthesize an extracellular matrix containing unique type X collagen [14]. In summary, chondrocyte hypertrophy is the key to activate osteocyte and necessary for osteogenesis differentiation. Longitudinal bone growth is the result of cell division and maturation in the proliferative region and maturation in the hypertrophic region [15]. This stage of chondrocytes is a collection of the proliferative, mature and hypertrophic zones [16].

In recent years, transposase accessible high-throughput sequencing (ATAC-seq) technology has made it possible to explore the epigenetic characteristics of various biological process in cattle. For example, dynamic changes in chromatin accessibility were characterized among four stages of bovine myoblast proliferation and differentiation [17]. Gao et al. reported potential regulation of chromatin dynamics in calf rumen epithelial tissues before and after weaning [18]. Many functional elements including promoters and enhancers were annotated in the differential peaks of adult and embryonic bovine muscle separately [19]. Underlying regulators in liver, muscle and hypothalamus, respectively, were identified by generating open chromatin profiles [20]. However, only limited evidence was available for chromatin accessibility during bovine cartilage development. To this end, we obtained the chromatin openness and transcription information of chondrocytes through self-built bovine chondrocyte bank. Through the integration of ATAC-seq and RNA-seq, we revealed the distribution regions of chromatin accessibility and potential transcriptional regulatory elements in fetal bovine chondrocyte. This study highlights the critical role in TFs function during cartilage development, which can provide a theoretical basis for expanding the understanding mechanisms regulating bovine body traits.

## 2. Materials and Methods

### 2.1. Animals

Cows were electrocuted and bled out. Three 90-day bovine fetuses were removed from the uterus of slaughtered pregnant cows.

### 2.2. Cell Isolation

The long bone cartilage tissues were extracted from the ends of a long bone. The perichondrium was peeled off, and then the cartilage was taken out with forceps. The cells were trypsinized with a 0.25% trypsin and washed with PBS twice. Then, a 0.1% type IV collagenase was added into tubes to digest.

### 2.3. ATAC-Seq Library Construction

ATAC-seq libraries were constructed with the TruePrep DNA Library Prep Kit V2 for Il-lumina (Vazyme, Nanjing, China). Briefly, 5 × 10^4^ chondrocytes were collected after di-gestion and resuspended in precool lysis buffer for 10 min. The lysed nucleus was re-suspended with a Tn5 transposase mixture and incubated at 37 °C for 30 min. The fragments were purified using the VAHTS DNA Clean Beads (Vazyme, Nanjing, China) and then amplificated for 16 cycles. Magnetic beads were used again to screen suitable fragments and the final libraries were quantitated using Qubit 4 Fluorometer (Invitro-gen, Singapore) and then sequenced on an Illumina HiSeq2500 platform following a PE150 strategy.

### 2.4. Total RNA Extraction and mRNA Library Construction

The mRNA (about 10^6^ chondrocytes) was purified from the total RNA with oligo-dT magnetic beads for three biological replicates, and then the RNA was fragmented with Mg^2+^. The RNA-seq library was constructed according to the following steps: reverse transcription, cDNA end repair, adding poly (A) tail. All libraries were sequenced on the HiSeq2500 platform according to the PE150 strategy.

### 2.5. ATAC-Seq Pre-Processing and Analysis

The sequencing quality, adapter content and duplication rates of the original reads from each ATAC-seq library were detected by FastQC(v.0.11.7) [21]. Trimming for the removal of Illumina adapters and low-quality sequences were performed by Trim Galore (v.0.6.3). The bovine genome reference (ARS-UCD1.2) and annotation were downloaded from the UCSC source database. After the bowtie2_index indices were built, the trimmed reads for each replicate were mapped to the ARS-UCD1.2 version of the bovine reference genome using Bowtie2 (v2.3.5) with default parameters. The reads were offset by +4 bp for the +strand and −5 bp for the −strand using deepTools software (v.3.3.0) [22]. All mapped reads for each individual replication were combined with samtools merge. MACS2 (v2.1.2) [23] was used to identify ATAC-seq peaks with the following parameters: –nomodel–shift-100–extsize 200 (the 5′ end extends to the 3′ end, and the sliding window is set to 200 bp). Individual peaks that occurred in at least two replicates and overlapped in more than 50% overlap in the two replicates were retained as candidate peaks [24]. Based on the gene functional elements such as promoter, 5′UTR, 3′UTR, exon and intron, ATAC-seq peaks were annotated with the annotatePeak function by R package ChIPseeker (v1.26.0) [25]. When the same peak fell on different components, we commented in the above order.

### 2.6. GWAS Enrichment Analysis in Peaks

As described previously, a total of 27,214 Holstein bulls with 2,048,052 imputed SNPs and available stature traits were included [26]. The original coordinates of the GWAS Summary Statistics were converted from UMD 3.1.1 to ARS-UCD1.2 using UCSC LiftOver tools [27]. Perl script (https://github.com/WentaoCai/GWAS_enrichment), URL (accessed on 20 April 2022) was used for the GWAS enrichment analysis to check whether SNP effects were more enriched in ATAC-seq peaks than in background area [28].

### 2.7. GO and KEGG Enrichment Analysis of Genes

Open chromatin regions were annotated to their nearest genes. GO enrichment analysis and KEGG pathway enrichment were carried out by the R package clusterProfiler [29]. The parameters were as follows: OrgDb = org.Bt.eg.db (genome-wide annotation for bovine of the GO database, which set the reference annotation package of the whole bovine genome), keyType = “ENTREZID” (EntrezGene ID), org = “bta” (organism of cow genome information), ont = “BP” (biological process), pAdjustMethod = “BH” (more strict multiple hypothesis test correction method), qvalueCutoff = 0.01.

### 2.8. Motif Enrichment and Footprinting Analysis

According to the position of the open chromatin region, the enrichment of the motifs corresponding to known TFs was performed using the findMotifsGenome.pl function of HOMER [30]. The significantly enriched transcription factor motifs were calculated by comparing the sequence set of the whole bovine genome motif library and target ATAC-seq peaks. TF-DNA binding prevents Tn5 transposases from cleavage in otherwise nucleosome-free region. HINT-ATAC was adapted to predict transcription factor footprints sites for open chromatin regions [31].

### 2.9. RNA-Seq Pre-Processing and Analysis

Raw sequencing fastq files were assessed for quality, adapter content and duplication rates with FastQC (v.0.11.7). The sequence reads for the removal of Illumina adapters and low-quality sequences were trimmed by Trim Galore (v.0.6.3). Trimmed reads were aligned to the ARS-UCD1.2 version of bovine reference genome using Tophat2 software (v.2.1.1) with the default setting. Quantification of gene expression counts were conducted by Hisat2 (v2.2.1) [32] and then normalized by Transcript per kilobase per million mapped reads (TPM) values. For downstream analysis, average gene expression TPM > 2 for the replicates were selected as expressed genes.

### 2.10. Visualization of ATAC-Seq and RNA-Seq Signal

BigWig files for the coverage tracks of ATAC-seq and RNA-seq were generated using the bamCoverag function of the deeptools (v.3.3.0) software. Visualization of the ATAC-seq and RNA-seq coverage track was conducted by Integrated Genomic Viewer (IGV 2.6.2).

## 3. Results

### 3.1. Quality Control and Alignment of ATAC-seq Data

After the paired-end sequencing data was filtered, 9.75 Gb of high-quality data was obtained. A preliminary analysis on the quality of the data was conducted. Raw sequences accounting for less than 0.5% of the total data volume were filtered in the quality control. We found that above 95% of clean reads were mapped to the bovine reference genome (bosTau9 version) (Table 1), which all meet the requirements of sequencing quality and further analysis. Evaluation of the library quality after quality control are shown in Appendix A. The fragments of chromatin open regions cut by Tn5 transposase are mainly divided into nucleosome-free region (<100 bp), nucleosome monomer (~200 bp), nucleosome dimer (~400 bp) and nucleosome trimer (~600 bp) [33]. Therefore, the length of fragments was periodically distributed with about 200 bp, which indicates that many fragments were protected by integer nucleosomes. Generally, the results of the reference sequence alignment analysis (Reads Mapping) and correlation between biological replicates further demonstrated the reliability of our ATAC-seq data.

### 3.2. Quality Control and Alignment of RNA-Seq Data

A total of 6.05 Gb of high-quality RNA-seq data was obtained after quality control. The mapping rates of three RNA-seq biological replicates were higher than 85% and the data quality satisfied the downstream analysis standard. In addition, Pearson’s correlation coefficient was 0.8 between biological replicates. Our RNA-seq data retained 10,962 genes with the expression levels greater than two (TPM > 2) in the three replicates.

### 3.3. Open Chromatin Regions Were Enriched at TSSs including Proliferation-Related Genes

We performed ATAC-seq to map the open chromatin accessible regions in fetal bovine chondrocytes. Three biological replicates and their combined data were named rep1, rep2, rep3, and merged peak, respectively. To establish a common peak set, the strict strategy for candidate peaks were adopted. Finally, a total of 9860 peaks were identified as shown in Figure 1a with an average length of 572 bp, accounting for 0.20% of the bovine genome. Examples of the ATAC-seq coverage track for genes related to cartilage development and body stature are shown in Figure 1b. The horizontal axis represents the genome coordinates, and the vertical axis represents the signal intensity. Our results indicated that the ATAC-seq signals were highly enriched at the transcription starting sites of chondrocyte proliferation-related marker genes *SOX9* and *RUNX2*. In addtion, we found that *XBP1* and *REEP6* with intense signals may be the potential chondrocyte proliferation-related genes that were rarely reported to be directly associated with cartilage development.

### 3.4. Genomic Distribution of Chromatin Accessibility Regions

Chromatin opening usually indicates that DNA with transcriptional regulatory elements is bound by transcriptional factors to regulate gene expressionand replication or transcription occurs [6]. To explore the regulatory function of the chromatin landscape in fetal bovine chondrocyte, we next annotated the genomic features of these open chromatin regions; 9686 of 9860 peaks were annotated on the genome. The priority position of peak distribution was enriched in the promoter (45.61%), followed by intergenic region (41.85%), gene downstream (1.17%), exon (0.88%) and 3′UTR (0.29%). In our study, ±3000 bp of the transcription start sites (TSSs) was selected as the promoter region. About 45% of peaks were located within 3 kb upstream and downstream of the TSS, and less than 42% of ATAC-seq peaks were located in distal intergenic regions (Figure 2). Therefore, the majority of the peaks located in the promoter regions were defined as the core promoter regions for subsequent analysis.

### 3.5. Function Enrichment Analysis of the Chondrocyte Open Chromatin Regions nearby Genes

In order to characterize the biological processes regulated by these open chromatin regions, we performed GO and KEGG enrichment analyses with the nearest genes. The corresponding GO terms and the functional categories are shown in Figure 3a. GO terms and KEGG pathways with *p* ≤ 0.05 were considered significantly enriched. Our results of GO terms demonstrated a highly significant enrichment of the chondrocyte-related processes such as peptide synthesis, amide synthesis, translation and regulation. The results of KEGG pathway enrichment showed a highly significant enrichment in the endoplasmic reticulum protein processing, ribosome, nucleocytoplasmic transport and other pathways (Figure 3b). These functional analyses showed that chondrocyte proliferation-related open chromatin regions may be closely associated with the protein synthesis and cell cycle processes.

### 3.6. Enrichment of the Cartilage-Development-Related TFs

TFs combine specific DNA sequences to activate or inhibit the transcription of DNA, and modulate the gene expression to control a series of key cellular processes [34]. In our data, the motif enrichment analysis was performed on the ATAC-seq-peak-related genes. The total target sequences are 9686, and the total background sequences are 34,355. The top 50 (*p*-value < 0.01) enriched motifs are shown in Figure 4a. The TFs specifically related to cartilage development were significantly enriched, such as NF-Y, ATF3, FOSL2, ATF2 and FOS. The *p*-value generated by TFs enrichment is sorted out, and the corresponding significance is listed in descending order as follows: NF-Y (1 × 10^−616^), ATF3 (1 × 10^−317^), FOSL2 (1 × 10^−300^), ATF2 (1 × 10^−260^) and Fos (1 × 10^−106^).

### 3.7. Footprint Verification of Significantly Enriched TFs

Footprint visually displays that the ATAC-seq signal near the TF motif in the 200 bp genome range, read coverage suddenly drops within peak regions of high coverage [35] (Figure 4b). The principle of this analysis is to protect the TF binding region from being targeted by Tn5 transposase, which is used to label accessible chromatin [36]. In addition, TF footprint analysis showed that the area of reduced accessibility is related to the combination of NF-Y, ATF3, FOSL2, ATF2 and FOS. On the contrary, the BATF was located at the top 10 of motif enrichment, but the footprint curve was disordered. It showed that TFs were inactive near DNA and did not bind to DNA.

### 3.8. GWAS Enrichment Analysis of Chondrocytes Open Chromatin Regions

To check whether SNPs effect was more abundant in the ATAC-seq peaks than those in the background area, we obtained the result of random sampling. The candidate SNPs were randomly sampled for 10,000 times and the number of samples each time was also random. The proportion of falling in the peak was summarized and counted, and the *p* value was calculated by one-tailed test (Table 2). *p*-value was equal to 0.0008 (<0.05 is considered as significant enrichment). From a different viewpoint, the reliability of peak regions could be certificated.

## 4. Discussion

Some studies have reported that epigenome data were used to identify the developmental regulatory mechanisms of bovine tissues (such as skeletal muscle, liver and rumen). Chromatin open profiles have also been constructed in the study on growth traits in pig, chicken and other livestock and poultry, but chromatin openness during fetal bovine cartilage development has not yet been reported [20,37,38].

Chondrocyte differentiation is usually accompanied by the biological synthesis of various proteins. Generally, the genetic programming of cartilage undergoes a multistep process. Cell types experience a change consisting of mesenchymal condensation, highly enriched extracellular matrix (ECM) cells, and chondrocytes [39]. The significant involved KEGG pathways include endoplasmic reticulum protein processing, ribosomes, TGF-β signaling, PI3K-Akt signaling, etc. Notably, the main component of ECM is type II collagen. Therefore, it is necessary to synthesize a large number of chondrocyte-related proteins and enhance the transport efficiency. Ribosome, endoplasmic reticulum, Golgi apparatus and other organelles produce a marked effect jointly. On the other hand, endoplasmic reticulum stress causes osteoarthritis and other related diseases [40], so the correct folding of proteins in the endoplasmic reticulum is particularly important. This mechanism relies on the polypeptide structure and the intracellular network. After the secreted protein enters the endoplasmic reticulum, the molecular chaperone system functions. For example, molecular chaperone calreticulin (CRT) mainly interacts with secretory proteins released by ribosomes to keep the substrate in the endoplasmic reticulum and facilitate protein folding [41]. During chondrocyte cultured from the third day to the fifteenth day, type II and type X collagen, proteoglycan core protein and other characteristic markers of chondrocyte differentiation was expressed [42]. Overexpression of fibronectin FN1, in a TGF-β/PI3K/Akt pathway, is positively correlated with the increase in type II collagen, type I collagen and osteonectin levels, thus completing bone formation and chondrocyte differentiation, and promoting fracture healing [43]. Cell proliferation also depends on the four different stages of the cell cycle; TGF-β1 can accelerate the process of G1 to S phase and complete the differentiation of antler chondrocytes [44].

Amide biosynthesis and translation regulation were more prominent in GO enrichment. Glutamine has a versatile role in cell proliferation and metabolism, participating in biosynthetic precursors and tricarboxylic acid cycle [45]. It is also a kind of metabolic fuel that meets the increasing ATP required for cell growth. As one of the metabolic pathways in chondrocytes, glycolytic pathway is less efficient compared with others [46]. Glucose-derived carbons were diverted during this period. Glutamine just meets the requirements of high biosynthetic needs as well as complements glucose in glycolytic pathway [47]. Some experiments have found that histone acetylation needs to be completed in the process of amide synthesis of chondrocytes. Glutamine is synthesized by glutamate dehydrogenase dependent acetyl coenzyme A, which controls the expression of cartilage-related genes. In addition, transaminase-mediated aspartic acid synthesis supports chondrocyte proliferation and matrix synthesis [48]. The highly efficient translation regulation networks include post-transcriptional and post-translational modifications. Chabronova et al. reported that some sites resulted in an altered ribosome translation regulation [49]. The correct post-translational regulation also decided the stability and activity of the protein of *RUNX2* [50]. In the result of open chromatin regions, *XBP1* and *REEP6* with highly effective signals were regarded as potential genes. Mice with *XBP1* deficient caused a chondrodysplasia phenotype [51]. *REEP6* is related to the structure of the endoplasmic reticulum, which is consistent with the previously enriched KEGG pathway, so it is speculated that *REEP6* is related to the synthesis of collagen on the endoplasmic reticulum.

From the peak annotation result, we consider that these open chromatin regions near TSSs contain a large number of potential transcription factor binding sites. NF-Y, a heterotrimeric complex composed of NF-YA, NF-YB, and NF-YC, specifically binds to CCAAT sequences in eukaryotic promoters [52]. Transcription factor NF-Y was significantly enriched in the motif results, which is consistent with the previous research on mammalian mice. The proximal core promoter of the *SOX9* gene, closely related to NF-Y, contains a CCAAT box, which is a cis-regulatory element responsible for responding to BMP-2 [53,54]. BMP-2 protein is involved in histone modification and chromatin remodeling during mouse cartilage formation. Transcription factor NF-Y combined with histone acetyltransferase p300 to form NF-Y-p300 complex. The complex combined with the promoter region of *SOX9* to activate the *Sox9* gene expression [54]. EMSA experiments also showed that NF-Y could specifically bind the CCAAT box in vitro [54,55]. In conclusion, the expression of *SOX9* is activated by NF-Y binding to the *Sox9* promoter region, which explains the high footprint and mRNA expression of the transcription factor NF-Y significantly enriched by the motif.

Transcription factors Fos, Fra2, Jun and other transcription factors together constitute the activator protein 1 (AP1) transcription factor family [56] which participate in various biological processes such as proliferation and differentiation. Karreth et al. determined that the transcription factor FRA2 encoded by *Fosl2* is necessary for cartilage development [57]. Due to the lack of transcription factor FRA2, the conditioned knockout mice had reduced chondrocyte differentiation during the whole development process, delayed postnatal growth and osteogenesis in spinal cartilage. *Fosl2*^−/−^ mice impaired chondrocyte differentiation in embryos and newborns and decreased the deposition of extracellular matrix (ECM) [58]. Histomorphological microcomputed tomography analysis of *FOSL2* tg and control mice at two different time points (4 weeks and 3 months of age) showed that *FOSL2* tg mice had increased bone volume and bone surface area [59].

Both ATF2 and ATF3 transcription factors belong to the ATF family, and the members of the ATF-CREB family play an important role in cartilage development. Mice lacking ATF2 displayed achondroplasia, and ATF2 also regulated cell proliferation by targeting cyclin A [60]. ATF3 has been widely reported in the treatment of diseases, such as prevention of cardiac hypertrophy and fibrosis, neuronal axon regeneration after traumatic nerve injury, liver fibrosis treatment, and breast cancer target [44,61,62,63]. James et al. determined that the mRNA expression of ATF3 increased significantly during the differentiation process of independent small pieces of embryonic bud cells in mouse embryos [64]. ATF3 was also found in our motif enrichment and footprint analysis. It is well known that *SOX9* promotes cell division in the early stage of proliferation, and also inhibits chondrocyte hypertrophy [65]. ATF3 overexpression inhibits *SOX9* activity, which may also explain that *SOX9* is not at a high level of expression. In cytokine-induced MMP13-derived human chondrocytes, the expression of the transcription factor FOS was almost halved from 1 h to 1.25 h, further suggesting a transient nature expression of the cartilage-specific transcription factor [66]. 

## 5. Conclusions

The development of chondrocytes in the fetal stage has a great influence on the bone morphology and bovine stature after birth. In this study, we characterized the chromatin accessibility of chondrocyte proliferation using fetal bovine long bone derived from chondrocytes. We identified the significant KEGG pathways and GO Terms related to cartilage development. Then, we identified several candidate TFs (NF-Y, ATF3, FOSL2, ATF2 and FOS) that might participate in the biological process of chondrocytes using motif enrichment and foot printing analysis. The results provide a theoretical basis for explaining the regulatory mechanisms of bovine fetal chondrocyte development.

## Figures and Tables

**Figure 1 animals-13-01875-f001:**
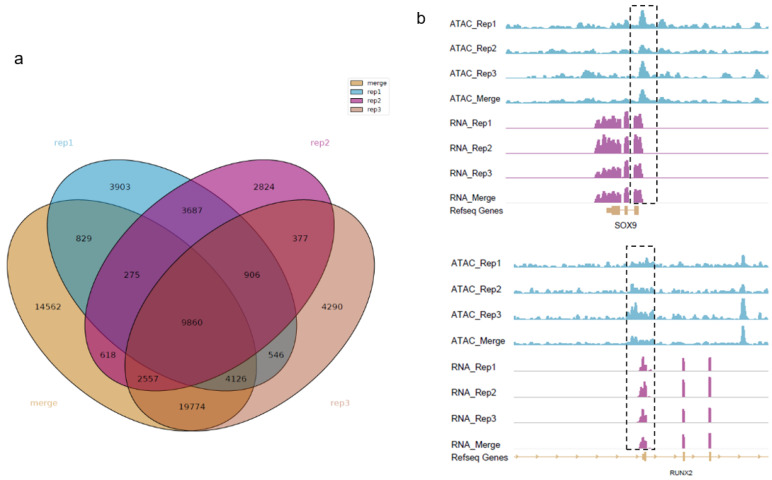
Venn diagram of calling peaks by MACS2 and IGV visualization. (**a**) A total of 9860 peaks are identified by grouping overlap call peaks for every two samples; (**b**) The location on the genome and peak map of genes *SOX9* and *RUNX2.* The dashed box shows the enriched signals of genes.

**Figure 2 animals-13-01875-f002:**
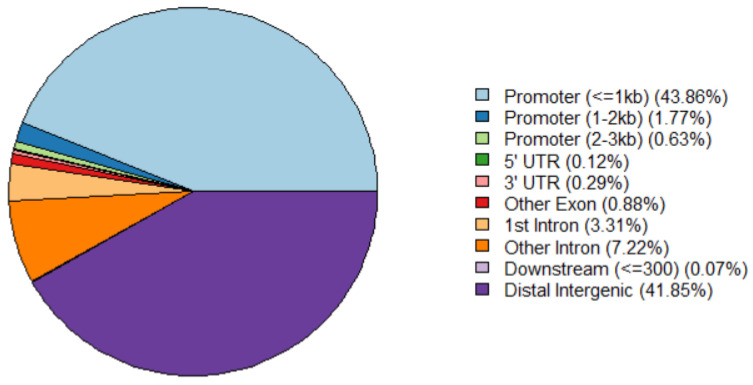
The annotation of ATAC-seq peak regions on the bovine genome.

**Figure 3 animals-13-01875-f003:**
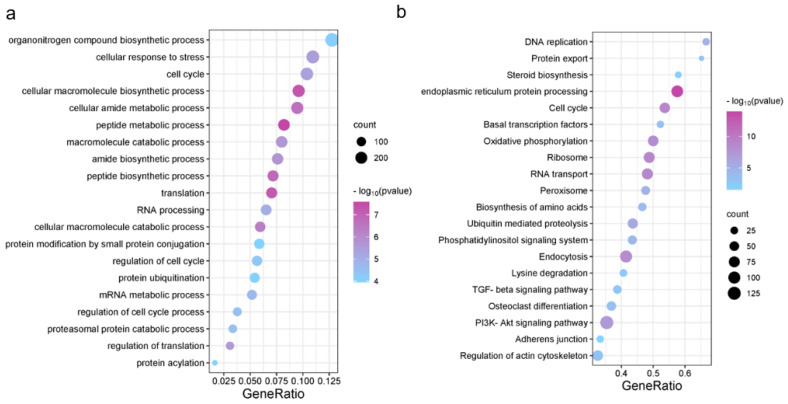
The results of GO and KEGG enrichment analysis. The horizontal axis shows the gene ratio. The vertical axis shows the biological process in GO or KEGG. (**a**) GO functional enrichment; (**b**) KEGG pathway enrichment.

**Figure 4 animals-13-01875-f004:**
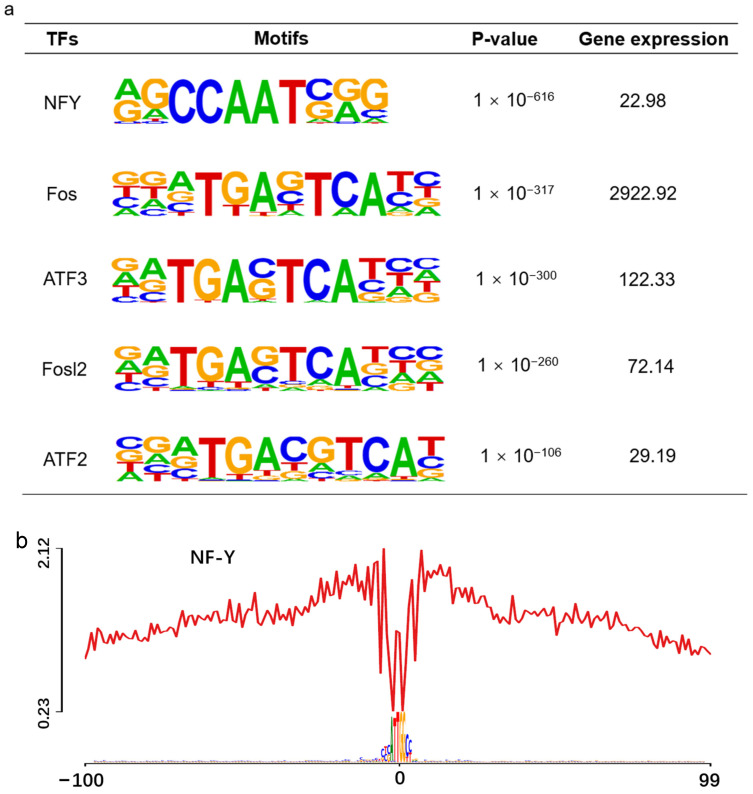
Motif enrichment and footprint analysis integrating ATAC-seq and RNA-seq data. (**a**) Enriched motifs and predicted TFs NF-Y, ATF3, FOSL2, ATF2 and FOS, *p*-value and mRNA expression; (**b**) The horizontal axis shows the distance from motif center. The vertical axis shows the number of reads (*n* = 765, 388, 577, 73, 238, and 2, respectively).

**Table 1 animals-13-01875-t001:** Summary of ATAC-seq and RNA-seq data after quality control.

Samples	Raw Reads	Clean Reads	Mapped Reads	Mapped Ratio (%)
ATAC-rep1	62,112,724	62,062,362	59,541,648	95.94
ATAC-rep2	72,352,432	72,297,570	69,924,312	96.72
ATAC-rep3	89,758,598	89,644,200	87,063,194	97.12
RNA-rep1	24,840,774	24,838,052	21,561,792	86.80
RNA-rep2	22,659,377	22,656,203	19,849,614	87.60
RNA-rep3	23,229,395	23,227,896	87,063,194	87.50

**Table 2 animals-13-01875-t002:** Summary of GWAS enrichment analysis in peaks.

Stature SNPs	*p*-Value	Fold
2,048,052	0.0008	2.32

## Data Availability

All relevant datasets generated in this study have been deposited in a NCBI SRA database with accession PRJNA952344.

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
