# Peer review of "Characterization of Chromatin Accessibility in Fetal Bovine Chondrocytes"

_animals, 2023, doi:10.3390/ani13111875_

Round 1

Reviewer 1 Report

This paper is not well written. The integration analysis of ATAC-seq and RNA-seq was included in the abstract, but RNA-seq was not presented in the results. Even though RNA-seq was presented in the form of IGV in the materials and methods, there were only the ATAC-seq (req1, req2, req3 and merge) in the figure 2.

The paper only shows the ATAC-seq and RNA-seq in one state, and does not compare the data of ATAC-seq and RNA-seq in two states, so this paper elaborated is very simple. The manuscript need to be major revised.

Line49-50: The sentence “The majority of TFs bind to open chromatin surveyed from the ENCODE project and chromatin accessibility status reflects TF binding information[6].” revise as “TF binding information is reflected in chromatin accessibility status, as surveyed from the ENCODE project [6].”

Line64-72: What relationship between the whole paragraph and the topic of the article. “In summary, chondrocyte hypertrophy is the key to activate osteocyte and necessary for osteogenesis differentiation.” The sentence shown the important of chondrocyte hypertrophy, and logically, chondrocyte hypertrophy or chondrocyte no hypertrophy should show as experimental material by ATAC-seq and RNA-seq in next paragraph. Why the experimental material is fetal bovine chondrocyte? What’s the relationship?

Line 161-174: This content can be placed in an attachment.

Line 175-180: This content can be placed in an attachment.

Line 185: “3.3. Peak Callingchange to the conclusion of the paragraph, such as ATAC-seq signals enriched at the nearby chondrocyte proliferation-related genes.  

“3.4. Peak Annotation”, “3.5. GO and KEGG Enrichment Analysis of Annotated Genes”, “3.6. Motif Enrichment” and “ 3.7. Footprint Verification” should also be revised.

Line 189: 9,686 or 9860?

Lin223: The paragraph makes a conclusion shown that the relationship between GO, KEGG pathway and chondrocyte.

Line 230: delete the “blank”.

Line 257: 0.008 or 0.0008?

Figure 1,2,4,5 is too blurred, please revise them.

This paper is not well written.

Author Response

Dear reviewer,

We thank for your comments and suggestions on our paper, ‘Characterization of Chromatin Accessibility in Fetal Bovine Chondrocytes’ A point by point response to each comment is presented below in red font. What’ more, we have added RNA-seq (req1, req2, req3 and merge) of IGV in the figure 2. And in Figure 5, there are also calculation results for expression 

Comment 1:

Line49-50: The sentence “The majority of TFs bind to open chromatin surveyed from the ENCODE project and chromatin accessibility status reflects TF binding information[6].” revise as “TF binding information is reflected in chromatin accessibility status, as surveyed from the ENCODE project [6].”

Response: Thank you for the suggestion. The sentence has been revised.

Comment 2:

Line64-72: What relationship between the whole paragraph and the topic of the article. “In summary, chondrocyte hypertrophy is the key to activate osteocyte and necessary for osteogenesis differentiation.” The sentence shown the important of chondrocyte hypertrophy, and logically, chondrocyte hypertrophy or chondrocyte no hypertrophy should show as experimental material by ATAC-seq and RNA-seq in next paragraph. Why the experimental material is fetal bovine chondrocyte? What’s the relationship?
Response: Thanks for your comments. As suggested by the reviewer, we organized this part: different regions cannot be distinguished when dividing cells, and dedifferentiation will occur during in vitro culture. Longitudinal bone growth is the result of cell division and maturation in the proliferative region and maturation in the hypertrophic region[1]. And this stage of the chondrocytes is a collection of the proliferative, mature and hypertrophic zones[2]. At last, fetal bovine chondrocyte was chosen as experimental material. And this part can refer to the paper Mechanisms of bone development and repair. In order to make the logic of the manuscript more complete, we also added some content to the manuscript.

Comment 3&4:

Line 161-174: This content can be placed in an attachment.

Line 175-180: This content can be placed in an attachment.

Response:Thank you for the suggestion. After consideration, we also think that it is not appropriate to put figure 1 in the text, so we put them in the attachment called Figure S1.

Comment 5:

Line 185: “3.3. Peak Calling” change to the conclusion of the paragraph, such as ATAC-seq signals enriched at the nearby chondrocyte proliferation-related genes. 

Response:Thank you for the suggestion. The “3.3. Peak Calling” was changed to: ‘Open Chromatin Regions were enriched at TSSs including Proliferation-related Genes’.

“3.4. Peak Annotation”, “3.5. GO and KEGG Enrichment Analysis of Annotated Genes”, “3.6. Motif Enrichment” and “ 3.7. Footprint Verification” should also be revised.

Response:’3.4. Peak Annotation’ was changed to Genomic Distribution of Chromatin Accessible Regions’; ‘3.5. GO and KEGG Enrichment Analysis of Annotated Genes’ was changed to ‘3.5. Function Enrichment analysis of the Chondrocyte Open Chromatin Regions nearby genes’; ‘3.6. Motif Enrichment’ was changed to ‘Enrichment of the Cartilage-Development-Related TFs’; ‘3.7. Footprint Verification’ was changed to ‘Footprint Verification of Significantly Enriched TFs’. 3.8 was changed to‘GWAS Enrichment Analysis of Chondrocytes Open Chromatin Regions’

Comment 6:

Line 189: 9,686 or 9860?

Response: Sorry for the confusion. We have replaced ‘9,686’ with ‘9,860’ on Line 189.

Comment 7:

Lin223: The paragraph makes a conclusion shown that the relationship between GO, KEGG pathway and chondrocyte.

Response: Thanks for your comments. We have added a conclusion in the end of this section as following: ‘These functional analyses showed chondrocyte proliferation-related open chromatin regions may be closely associated with the protein synthesis in cartilage development.’

Comment 8:

Line 230: delete the “blank”.

Response: Thanks for your comments. “blank” has been removed.

Comment 9:

Line 257: 0.008 or 0.0008?

Response: Thank you for indicating this error. The result of the calculation is 0.0008.

We have revised this in our manuscript.

Comment 10:

Figure 1,2,4,5 is too blurred, please revise them

Response: Thanks for your suggestion. We have re-uploaded a clearer image.

  1. Villemure, I.; Stokes, I. A., Growth plate mechanics and mechanobiology. A survey of present understanding. J Biomech 2009, 42, (12), 1793-803.
  2. Petty, R. E., Chapter 2 - Structure and Function. In Textbook of Pediatric Rheumatology (Seventh Edition), Petty, R. E.; Laxer, R. M.; Lindsley, C. B.; Wedderburn, L. R., Eds. W.B. Saunders: Philadelphia, 2016; pp 5-13.e2.

Reviewer 2 Report

Zhang et al characterize the accessible chromatin regions in chondrocytes of cattle, and added to it the mRNA of the same chondrocytes.

They analyzed the long bone cartilage tissues of three bovine fetuses and performed aTaC seq and RNA-seq to characterize the gene expression levels and chromatin accessibility profile in bovine chondrocytes. The technical part of collecting the cells and the bioinformatics work was done in a great way. However, it is hard to conclude regarding the comparison between the open regions to the mRNA that is expressed in a one-time point. It would be great if the authors will compare their recent data with available data on adult bones or any other published work on bovine chondrocytes, or different chondrocytes' bovine stages, to get a better picture of the open chromatin regions and the mRNA expression of chondrocytes in different stages. The motif enrichment analysis is great and can help in putting the groundwork for future experiments in the development of chondrocytes and the transitions between their different stages. However, it is important to define the exact stage of the chondrocytes that were taken from the bovine fetuses.

Major comments:

1. It's unclear to the reader what are the ages of the bovine fetuses. Indeed, in the abstract (line 15), the authors mention 90 days for the bovine ages. However, the ages of the embryonic animals should be mentioned in the methods section. Moreover, explaining why the authors decided on this specific embryonic stage would be helpful for the reader. It would also be helpful if the authors could elaborate more on the chondrocytes' development in the long bone tissue. Did the authors take a specific place in the long bones? Or did they process all the long bones and extract the cells from the complete region?  

2.    Total mRNA was purified from the total RNA (line 103) did the authors take it from the same samples?

3.    Sox9 and Runx2 are great candidates that will be open. Did the authors find new genes that participated in the chondrocytes' growth or other interesting unknown regions?

4.    The authors took only a one-time point of chondrocyte proliferation, therefore it’s hard to compare the mRNA to the aTaC seq data for different gene expressions or different accessible regions. Comparing this one-time point to adult bone or older fetuses would be helpful to conclude more robust results regarding the aTaC data and the mRNA data.

Minor comments:

1.    Please add in the methods a section regarding the isolation of the cells from the long bones. What was the bovine development age? What parts were taken from the bone, and how many cells were extracted from each sample for aTaC sew and mRNA seq.

2.     In Figure 2b the authors should mark the exact place of the Sox9 and Runx2 open regions

The English quality is good and the paper is readable 

Author Response

Dear reviewer,

We thank for your comments and suggestions on our paper, ‘Characterization of Chromatin Accessibility in Fetal Bovine Chondrocytes’ A point by point response to each comment is presented below in red font.

Comment 1:

Major comments:

It's unclear to the reader what are the ages of the bovine fetuses. Indeed, in the abstract (line 15), the authors mention 90 days for the bovine ages. However, the ages of the embryonic animals should be mentioned in the methods section. Moreover, explaining why the authors decided on this specific embryonic stage would be helpful for the reader. It would also be helpful if the authors could elaborate more on the chondrocytes' development in the long bone tissue. Did the authors take a specific place in the long bones? Or did they process all the long bones and extract the cells from the complete region? 

Response:Thank you for the suggestion. We have added the 90 days for the bovine ages in the methods section. During bovine fetal development, most of the organs differentiated at the 90-day of gestation including long bones. At three months of age, long bones developed rapidly under the maternal supply of nutrients. Also, long bones of 90-day fetus are soft and easy to sample. In addition, the experimental material mentioned in the manuscript is the cartilage tissue located at both ends of the long bone. First, the perichondrium is peeled off, and then the cartilage is taken out with forceps. This part of the chondrocytes contains the proliferative, mature and hypertrophic zones. The chondrocytes' development in the long bone tissue proceeds as follows: The proliferative zone produces new chondrocytes by mitosis, replacing the apoptotic chondrocytes at the diaphysis. Longitudinal bone growth is the result of cell division and maturation in the proliferative region and maturation in the hypertrophic region. And this part can refer to the paper Mechanisms of bone development and repair.

Comment 2:

Total mRNA was purified from the total RNA (line 103) did the authors take it from the same samples?

Response:Thanks for your comments. In manuscript, we described ‘The mRNA was purified from the total RNA with oligo-dT magnetic beads for three biological replicates’. We didn’t take it from the same samples. RNA was extracted from three fetal bovine separately as three biological replicates for analysis, and then three sets of transcriptome data were analyzed.

Comment 3:

Sox9 and Runx2 are great candidates that will be open. Did the authors find new genes that participated in the chondrocytes' growth or other interesting unknown regions?

Response:Thank you for the suggestion, we have found the chromatin open regions in XBP1 (chr17:68052009-68057008) and REEP6 (chr7: 43,899,571-43,901,428). Cameron et. generated and characterized mice in which XBP1 was functionally inactivated by deletion of exon 2 in cartilage (Xbp1CartΔEx2). Cartilage-specific ablation of XBP1 activity caused a chondrodysplasia phenotype involving shortening of endochondral bones[1]. REEP6 has not been reported to be directly related to cartilage growth, but this gene is related to the structure of the endoplasmic reticulum, which is consistent with the previously enriched KEGG pathway, so it is speculated that REEP6 is related to the synthesis of collagen on the endoplasmic reticulum.

Comment 4:

  1. The authors took only a one-time point of chondrocyte proliferation, therefore it’s hard to compare the mRNA to the aTaC seq data for different gene expressions or different accessible regions. Comparing this one-time point to adult bone or older fetuses would be helpful to conclude more robust results regarding the aTaC data and the mRNA data.

Response: Thanks for your comments. As suggested by the reviewer, further efforts need to be directed toward comparing this one-time point to adult bone or older fetuses. Our study was limited by available public data of bovine chondrocytes experiment from prior study. In the future, we will pay more attention to published research and try our best to refine our study.

Minor comments:

Comment 5:

  1. Please add in the methods a section regarding the isolation of the cells from the long bones. What was the bovine development age? What parts were taken from the bone, and how many cells were extracted from each sample for aTaC sew and mRNA seq.

Response:Thank you for the suggestion. 5 × 104 chondrocytes were extracted from each sample for ATAC-seq and 106 chondrocytes were extracted from each fetal bovine for RNA-seq. For example, ATAC_Rep1 and RNA-Rep1 are from the same sample. And we have added two sections in Materials and methods called ‘2.1 Animals’ and ‘2.2 Cell Isolation’.

Comment 6:

  1. In Figure 2b the authors should mark the exact place of the Sox9 and Runx2 open regions

Response:Thank you for your suggestion. We have marked the exact place of the Sox9 and Runx2 open regions by dashed box in figure 2b.

  1. Cameron, T. L.; Gresshoff, I. L.; Bell, K. M.; Piróg, K. A.; Sampurno, L.; Hartley, C. L.; Sanford, E. M.; Wilson, R.; Ermann, J.; Boot-Handford, R. P.; Glimcher, L. H.; Briggs, M. D.; Bateman, J. F., Cartilage-specific ablation of XBP1 signaling in mouse results in a chondrodysplasia characterized by reduced chondrocyte proliferation and delayed cartilage maturation and mineralization. Osteoarthritis Cartilage 2015, 23, (4), 661-70.

Round 2

Reviewer 2 Report

The authors answered well to most of the comments.

I would try to add a graphical abstract or another sub-figure in Figure 1, to present the flow of the experiment, and from where did they take the cells ( picture of long bones and the specific sites where they took the cells from). I would put 

Author Response

Dear reviewer,

We thank for your comments and suggestions on our paper, ‘Characterization of Chromatin Accessibility in Fetal Bovine Chondrocytes’ A point by point response to each comment is presented below in red font.

Comment: 

I would ask the writers to add a graphical abstract that describes the flow and method of the cell collections. Picture or drawing of the long bones with the specific taken sites, the number of cells, etc. It will help the reader to understand the design of this paper. 

Response:Thank you for the suggestion. We have added the graphical abstract to help the reader to understand the design of this paper.
